# A systematic review and meta-analysis of indoor bioaerosols in hospitals: The influence of heating, ventilation, and air conditioning

**Rongchen Dai[1‡], Shan Liu[2‡], Qiushuang Li[2], Hanting Wu[1], Li Wu[2], Conghua Ji[1,2]***

**1** School of Public Health, Zhejiang Chinese Medical University, Hangzhou, Zhejiang Province, China,
**2** Center of Clinical Evaluation and Analysis, The First Affiliated Hospital of Zhejiang Chinese Medical University, Hangzhou, Zhejiang Province, China

‡ RD and SL are contributed equally to this work and should be considered co-first authors
* jchi2005@126.com

**Data Availability Statement:** All relevant data are within the manuscript and its Supporting Information files.

## Abstract

### Objectives

To evaluate (1) the relationship between heating, ventilation, and air conditioning (HVAC) systems and bioaerosol concentrations in hospital rooms, and (2) the effectiveness of laminar air flow (LAF) and high efficiency particulate air (HEPA) according to the indoor bioaerosol concentrations.

### Methods

Databases of Embase, PubMed, Cochrane Library, MEDLINE, and Web of Science were searched from 1st January 2000 to 31st December 2020. Two reviewers independently extracted data and assessed the quality of the studies. The samples obtained from different areas of hospitals were grouped and described statistically. Furthermore, the meta-analysis of LAF and HEPA were performed using random-effects models. The methodological quality of the studies included in the meta-analysis was assessed using the checklist recommended by the Agency for Healthcare Research and Quality.

### Results

The mean CFU/m$^3$ of the conventional HVAC rooms and enhanced HVAC rooms was lower than that of rooms without HVAC systems. Furthermore, the use of the HEPA filter reduced bacteria by 113.13 (95% CI: -197.89, -28.38) CFU/m$^3$ and fungi by 6.53 (95% CI: -10.50, -2.55) CFU/m$^3$. Meanwhile, the indoor bacterial concentration of LAF systems decreased by 40.05 (95% CI: -55.52, -24.58) CFU/m$^3$ compared to that of conventional HVAC systems.

### Conclusions

The HVAC systems in hospitals can effectively remove bioaerosols. Further, the use of HEPA filters is an effective option for areas that are under-ventilated and require additional protection. However, other components of the LAF system other than the HEPA filter are not conducive to removing airborne bacteria and fungi.

**Funding:** This work was supported by the key R & D projects from the Department of Science and Technology of Zhejiang Province [NO.2020C03126] and the Administration of Traditional Chinese Medicine of Zhejiang Province [NO.2017ZZ007], the People's Republic of China.

**Competing interests:** The authors have declared that no competing interests exist.

**Abbreviations:** COVID-19, coronavirus disease 2019; HVAC, heating, ventilation, and air conditioning; LAF, laminar air flow; CFU, colony forming units; CFU/m3, colony forming units per cubic meter; WHO, World Health Organisation; IAQ, indoor air quality; PRISMA, Preferred Reporting Items for Systematic Reviews and Meta-Analyses; SD, standard deviation; CI, confidence interval; TAF, turbulent air flow; ACH, air change per hour; HEPA, high efficiency particulate air; SSIs, surgical site infections.

## Limitation of study

Although our study analysed the overall trend of indoor bioaerosols, the conclusions cannot be extrapolated to rare, hard-to-culture, and highly pathogenic species, as well as species complexes. These species require specific culture conditions or different sampling requirements. Investigating the effects of HVAC systems on these species via conventional culture counting methods is challenging and further analysis that includes combining molecular identification methods is necessary.

## Strength of the study

Our study was the first meta-analysis to evaluate the effect of HVAC systems on indoor bioaerosols through microbial incubation count. Our study demonstrated that HVAC systems could effectively reduce overall bioaerosol concentrations to maintain better indoor air quality. Moreover, our study provided further evidence that other components of the LAF system other than the HEPA filter are not conducive to removing airborne bacteria and fungi.

## Practical implication

Our research showed that HEPA filters are more effective at removing bioaerosols in HVAC systems than the current LAF system. Therefore, instead of opting for the more costly LAF system, a filter with a higher filtration rate would be a better choice for indoor environments that require higher air quality; this is valuable for operating room construction and maintenance budget allocation.

## Introduction

Heating, ventilation, and air conditioning (HVAC) systems are widely used in hospitals to improve indoor personal comfort, relieve some temperature-related symptoms, and remove bioaerosols [1–3]. However, the coronavirus disease 2019 (COVID-19) pandemic has raised concerns that HVAC systems may increase the risk of airborne diseases if not well designed or properly managed [4–8]. Specifically, studies examining artificially generated aerosols indicated that SARS-CoV-2 is viable in aerosols, and that HVAC systems speed up and change the direction of indoor air flow [9]. As a result, some researchers suspected that HVAC systems may increase the risk of SARS-CoV-2 infection [6, 7], and suggested that poorly designed and managed HVAC systems are likely to provide convenient access to infectious diseases [10]. Therefore, research on the indoor bioaerosol of HVAC rooms to evaluate the advantages and risks of HVAC systems application in indoor environments may help guide the prudent use and management of HVAC systems, especially during the ongoing COVID-19 pandemic.

The laminar air flow (LAF) system is a system that provides unidirectional air flow in the operating room, but it is expensive to install and maintain and requires a lot of energy and ongoing technical maintenance [11]. Standard operating room ventilation filters air with the removal of 80–97% of particles > 5 μm. LAF systems equipped with high-efficiency particulate air (HEPA) filters remove 99.97% of particles > 0.3 μm that may reduce the risk of infectious disease transmissions since they remove particles and large droplets that may carry pathogens [12–15]. The WHO released guidelines in 2016 that suggested that LAF ventilation systems

should not be used to reduce the risk of SSIs (surgical site infections) for patients undergoing total arthroplasty surgery based on low quality of evidence [16]. Furthermore, a subsequent meta-analysis discovered that LAF systems have no apparent benefits over conventional turbulent ventilation in operating rooms when trying to reduce the risk of SSIs in total hip and knee arthroplasties or abdominal surgery [17]. LAF systems are currently used for high-risk septic/aseptic operation because some researchers and policy makers still believe that the LAF system is effective in removing bioaerosols anyway [18–22].

Bioaerosols are defined as airborne particles of liquid or volatile compounds that contain living organisms or that have been released from living organisms [23]. Indoor air quality (IAQ) is significantly affected by the concentration of bioaerosols, such as bacteria, fungi, viruses, and pollens. High bioaerosol concentration is associated with greater infectivity, sensitization, and toxicity [24]. At present, there is a study that roughly analysed the concentration and composition of indoor bioaerosols in hospitals, as well as their correlation with HVAC systems [25]. According to Stockwell et al., the indoor colony forming units (CFU) concentration values measured had a large fluctuation range. Moreover, as most of the data were from observational studies, there were many interference factors, leading to poor comparability. Therefore, we decided to expand the scope of retrieval and design stricter inclusion criteria. Furthermore, we designed a combination of statistical description and meta-analysis methods in the protocol [26]. We conducted descriptive statistics on all types of studies, including single group studies, and then selected and compared a qualified experimental group with a control group for meta-analysis.

Our preliminary research objective was to evaluate the effect of HVAC systems on the IAQ of hospitals by determining the concentration of bioaerosols. Furthermore, this study aimed to determine whether the LAF systems and HEPA filters used in hospitals effectively influence these bioaerosol concentrations.

## Materials and methods

The present systematic review and meta-analysis were performed according to a protocol designed a priori following recommendations set by the Preferred Reporting Items for Systematic Reviews and Meta-Analyses (PRISMA) guidelines. The present work has been registered at the International Prospective Register for Systematic Reviews, identification code (CRD42020223461) [26].

### Search strategy

A comprehensive search for relevant studies was conducted in the following electronic databases: Embase, PubMed, Cochrane Library, MEDLIN, and Web of Science Core Collection (search query listed in S1 File). Since older HVAC systems may not be technically comparable to modern ventilation systems, researches published before 1st January 2000 were excluded. According to the databases mentioned above and the limiting conditions, we searched a total of 27,610 manuscripts before removing the duplicates. We also screened the reference lists of these literature reviews for further eligible publications.

### Selection criteria

The selection was conducted by three of our team's evaluators (R.C.D, S.L, C.H.J) using the EndNote X9.2 software. Removal of duplicates was then performed. After scrutinizing the inclusion and exclusion criteria, two evaluators (R.C.D, S.L) independently classified all studies as either 'Yes', 'No', or 'Unclear'. An article was formally included (or excluded) only when both evaluators agreed to that decision; otherwise, the three evaluators (R.C.D, S.L, C.H.J)

voted on the matter. There were two stages to this process: a preliminary screening of the title and summary for potentially relevant studies and a detailed screening of the full text. All three evaluators were needed for both stages.

The inclusion criteria for the studies were as follows: (1) studies published (in English) between 1st January 2000 and 31st December 2020, (2) studies wherein air sampling was undertaken indoors in a hospital using the active sampling method, (3) studies that explicitly descried the characteristics of the HVAC system, and (4) studies that quantitatively reported the results in CFU per cubic meter (CFU/m$^3$).

Journal articles were excluded based on the following criteria: 1) the data provided were related only to specific microorganisms (e.g. results limited to legionella); or 2) they were non-original articles (e.g. reviews); or 3) only computational fluid dynamics were used for numerical simulations; or 4) only hospital surfaces were sampled; or 5) lack of classification information (e.g. unable to determine the exact sampling area); or 6) the mean or standard deviation of CFU/m$^3$ was missing and could not be calculated by formulas in S1 Table [27].

## Data extraction

The following information was extracted from each study: the basic information of the articles, conditions for incubating microorganisms after sampling, characteristics of HVAC systems, and concentrations of indoor bioaerosols.

The indoor sampling sites at the hospitals were categorized into publicly accessible areas, inpatient facilities, and restricted areas. Restricted areas/rooms were those that access to which was restricted or required wearing personal protective equipment (e.g. operating room; intensive care unit; haemodialysis room). Ventilation methods were classified as natural ventilation, conventional HVAC systems, and enhanced HVAC systems. Natural ventilation was defined as ventilation that provided air flow by means of opening doors or windows and without an HVAC system. Conventional HVAC systems were defined as HVAC systems (1) without LAF, (2) with a filtration rate of ≤95%, and (3) with an ACH (air change per hour) < 15 exchanges per hour. Enhanced HVAC systems were defined as HVAC systems (1) that equipped with LAF, (2) that have a filtration rate of >95%, or (3) ACH ≥15 exchanges per hour.

In the case of a missing mean or standard deviation (SD) data, the formulas in S1 Table were used for conversion [27]. After the conversions, there were still multiple data with consistent classification. For example, some articles compared working and non-working states [28] or compared different medical operation procedures [29]. These classifications were not within our research scope, and so we used the Review Manager 5.4.1 Calculator to merge the results of the same classification in the qualitative analysis. Moreover, the sample size could not be determined, we took the minimum number that could be determined as the sample size (e.g., number of rooms sampled or number of surgical cases). We also adjusted the classification of total viable count in some articles according to the type of medium [30–32]. Beyond that, a small number of articles only stated that air conditioning was not used [33], and we could not accurately judge whether only doors and windows were used for ventilation in that room (or whether the air conditioning had simply been turned off). Because the design structure layout of rooms completely relying on natural ventilation must be different from that of air-conditioned rooms, we divide the data into two groups according to the author's description: those with natural ventilation and those without HVAC systems.

One reviewer (R.C.D) extracted the data while a second reviewer (H.T.W) ensured the data extraction was accurate and complete. The reviewers discussed all data discrepancies to achieve a consensus, and any uncertainties were resolved by the team members.

## Methodological quality assessment

The methodological quality of the studies included in the meta-analysis was assessed using an 11-item checklist recommended by the Agency for Healthcare Research and Quality for cross-sectional/prevalence study quality [34]. An item would be scored '0' if the response was 'NO' or 'UNCLEAR', and '1' if it was 'YES'. This was conducted independently by two members of the team. These two reviewers discussed any discrepancies until resolved. The rating details can be found in S1 Appendix.

## Statistical analysis

Statistical description was carried out using SPSS 26 (IBM Corp) and the Wilcoxon signed rank test was used to compare differences according to the characteristics of the final data. Meta-analysis was performed through Stata/MP 14.0. The weighted mean difference and random effects model were used for each comparison. Sensitivity analyses were completed to test the robustness of our finding. Heterogeneity among the studies was tested using of the inconsistency index ($I^2$), and Begg's Test was employed to assess the occurrence of whether publication[35]. A P-value <0.05 was considered significant.

# Results

Fig 1 presents our study selection process. In total, 27,610 records were retrieved during our initial search. After screening the reviews, an additional 46 records from the review were included. Thereafter, 3492 records were excluded using EndNote's de-duplication tool. A total of 101 articles were assessed for eligibility after the selection of titles and abstracts. Of these articles, we excluded others in accordance with the above-mentioned exclusion criteria. Finally, 30 articles were included in the statistical description and 11 articles were included in the meta-analysis. The characteristics of articles included in the statistical description and meta-analysis are reported in Tables 1 and 2, respectively. The data extracted from the included articles are provided in S2 Table, and we annotated the converted data.

A total of 9336 samples were included in the statistical analysis, and the result of bacteria and fungi are presented in Tables 3 and 4, respectively. This included 4100 bacterial samples and 5236 fungal samples collected from different HVAC systems in various areas of the hospital. In total, 91.20% (n = 3739) and 98.82% (n = 3695) of the bacterial samples were from restricted areas and restricted areas with enhanced ventilation systems, respectively (Fig 2). Overall, 76.68% (n = 4015) and 75.87% (n = 3046) of the fungal samples came from inpatient facilities and inpatient facilities with conventional HVAC systems, respectively (Fig 3). In contrast, few extant studies drew samples from publicly accessible areas in hospitals such that only the fungal counts were available for analysis and all samples were taken from conventional HVAC conditions (n = 575).

The mean CFU/m³ of the conventional HVAC rooms (bacterial count: 217.69 ± 116.69 CFU/m³; fungal count: 37.33 ± 82.78 CFU/m³) and of the enhanced HVAC rooms (bacterial count: 35.94 ± 39.55 CFU/m³; fungal count: 9.46 ± 9.63 CFU/m³) were lower than the rooms without HVAC systems (bacterial count: 360.82 ± 164.40 CFU/m³; fungal count: 38.17 ± 101.36 CFU/m³). In all areas, the indoor mean bioaerosol concentrations of rooms without HVAC system, with conventional HVAC systems and enhanced HVAC systems decreased sequentially (Figs 4 and 5). The concentrations of bacteria and fungi in HVAC rooms in various areas of the hospitals are shown in the Tables 3 and 4.

In hospital environments using HVAC systems, we calculated the mean bioaerosol concentrations under the classifications of LAFand turbulent air flow (TAF) conditions, high ACH and low ACH conditions, and HEPA filter and other filter conditions. All of the results are

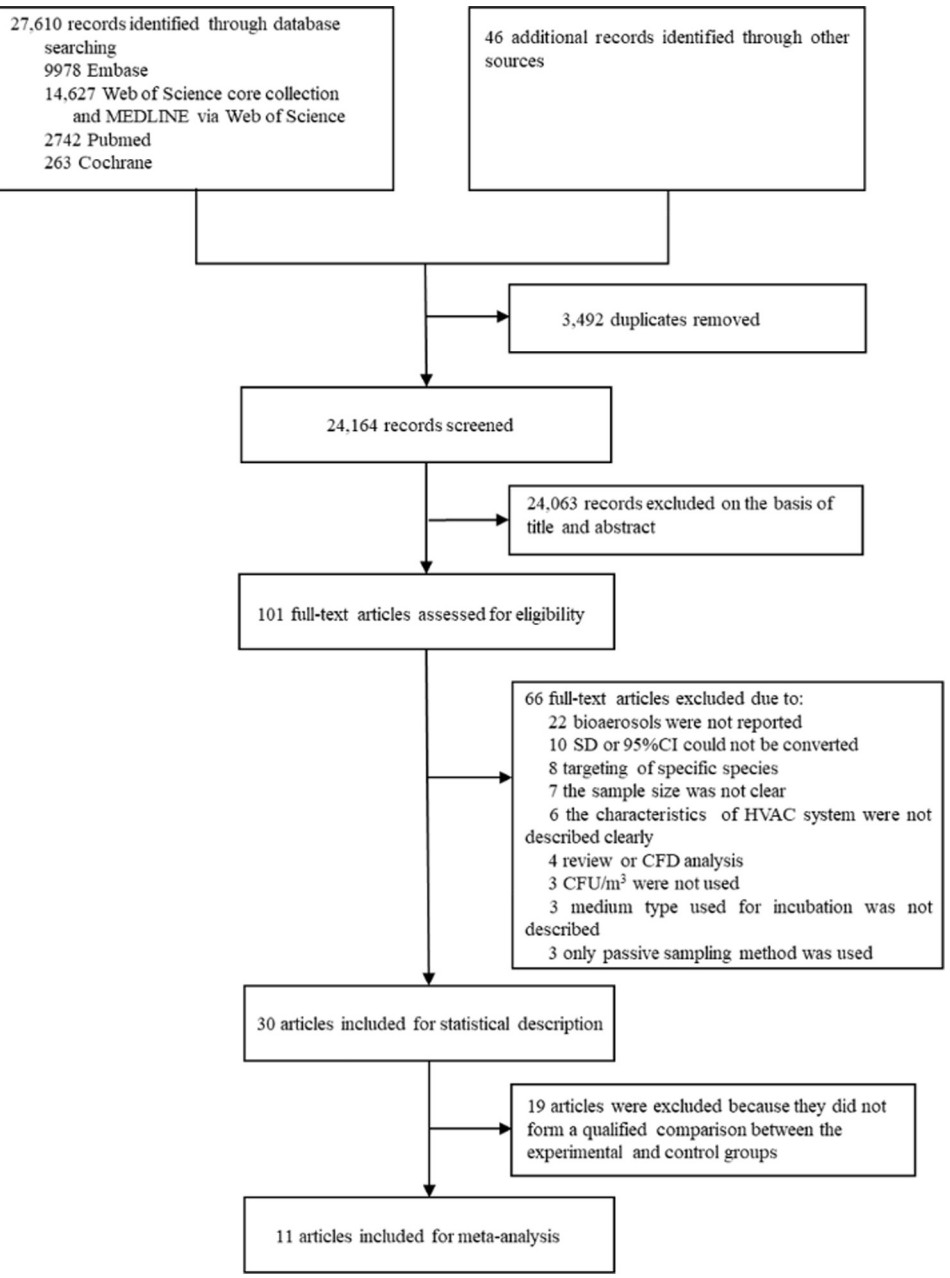

**Fig 1. Flow diagram of study selection.**

presented in Table 5. Conditions wherein HEPA filters were used (bacterial count: 36.90 ± 40.06 CFU/m$^3$; fungal count: 9.46 ± 9.63 CFU/m$^3$) showed lower mean bioaerosol concentrations than those wherein other filters were used (bacterial count: 57.83 ± 85.06 CFU/m$^3$; fungal count: 12.21 ± 14.22 CFU/m$^3$). High ACH conditions (bacterial count: 53.45 ± 47.15 CFU/m$^3$; fungal count: 22.96 ± 17.17 CFU/m$^3$) showed significantly lower mean counts of fungi than low ACH conditions (bacterial count: 58.05 ± 53.53 CFU/m$^3$; fungal count: 22.96 ± 17.17 CFU/m$^3$), while there were no significant differences (P = 0.175) in the mean counts of bacteria. For LAF systems, bacterial and fungal counts presented opposite results. The mean and SD of bacterial count in LAF conditions (bacterial count: 26.28 ± 29.78 CFU/

**Table 1. Characteristics of included studies.**

| Reference | Author | year | Microbiological count type | Hospital locations tested | Ventilation method | ACH | Filter type | Type of ventilation system |
|---|---|---|---|---|---|---|---|---|
| [36] | Fu Shaw et al. | 2018 | BC | restricted areas | enhanced HVAC system | 40 air changes of filtered air per hour | HEPA | LAF |
| [33] | Hansen et al. | 2005 | BC | restricted areas | conventional HVAC system | unclear | F7 or F9 | without LAF |
| | | | BC | restricted areas | without HVAC system | NA | NA | NA |
| | | | BC | restricted areas | enhanced HVAC system | unclear | HEPA | LAF |
| [28] | Kedjarune et al. | 2000 | BC | inpatient facilities | conventional HVAC system | unclear | unclear | unclear |
| [37] | Khawcharoenporn et al. | 2013 | FC | inpatient facilities | natural ventilation | NA | NA | NA |
| | | | FC | inpatient facilities | natural ventilation | NA | NA | NA |
| | | | FC | inpatient facilities | conventional HVAC system | unclear | unclear | unclear |
| | | | FC | inpatient facilities | conventional HVAC system | unclear | unclear | unclear |
| [38] | Wan et al. | 2011 | BC | restricted areas | enhanced HVAC system | 15–23 air changes per hour | HEPA | LAF |
| [39] | Stocks et al. | 2010 | BC | restricted areas | enhanced HVAC system | a minimum of 15 exchanges per hour | Varicell filter (95% effective at removing particles ≥0.3μm) | TAF |
| [40] | Sautour et al. | 2009 | FC | inpatient facilities | conventional HVAC system | unclear | Plasmair™ | unclear |
| [31] | Napoli et al. | 2012 | BC | restricted areas | enhanced HVAC system | unclear | HEPA | TAF |
| [41] | Perdelli, Sartini et al. | 2006 | FC | inpatient facilities | without HVAC system | NA | NA | NA |
| | | | BC | inpatient facilities | without HVAC system | NA | NA | NA |
| | | | FC | inpatient facilities | conventional HVAC system | 6 exchanges per hour | minimum efficiency reporting value (80% to 85% efficiency) | TAF |
| | | | BC | inpatient facilities | conventional HVAC system | 6 exchanges per hour | minimum efficiency reporting value (80% to 85% efficiency) | TAF |
| | | | FC | inpatient facilities | enhanced HVAC system | 6 exchanges per hour | HEPA | TAF |
| | | | BC | inpatient facilities | enhanced HVAC system | 6 exchanges per hour | HEPA | TAF |
| [42] | Pasquarella et al. | 2012 | BC | restricted areas | enhanced HVAC system | 15 air changes per hour | HEPA | unclear |
| | | | FC | restricted areas | enhanced HVAC system | 15 air changes per hour | HEPA | unclear |
| [43] | Ortiz et al. | 2009 | FC | restricted areas | enhanced HVAC system | unclear | HEPA | unclear |
| [44] | Napoli et al. | 2012 | BC | restricted areas | enhanced HVAC system | 19.3 air changes per hour | HEPA | TAF |
| [45] | Landrin et al. | 2005 | BC | restricted areas | enhanced HVAC system | 30 air changes per hour | HEPA | unclear |

(*Continued*)

**Table 1.** (Continued)

| Reference | Author | year | Microbiological count type | Hospital locations tested | Ventilation method | ACH | Filter type | Type of ventilation system |
|---|---|---|---|---|---|---|---|---|
| [30] | Cristina et al. | 2012 | BC | restricted areas | enhanced HVAC system | 20 efficacious air exchanges were carried out per hour | HEPA | TAF |
| [46] | Brun et al. | 2013 | FC | publicly accessible areas | conventional HVAC system | unclear | without HEPA | unclear |
| | | | FC | inpatient facilities | conventional HVAC system | unclear | without HEPA | unclear |
| | | | FC | inpatient facilities | enhanced HVAC system | unclear | HEPA | unclear |
| [4] | Bozic et al. | 2019 | BC | inpatient facilities | without HVAC system | NA | NA | NA |
| | | | FC | inpatient facilities | without HVAC system | NA | NA | NA |
| | | | BC | inpatient facilities | conventional HVAC system | unclear | unclear | unclear |
| | | | FC | inpatient facilities | conventional HVAC system | unclear | unclear | unclear |
| [32] | Albertini et al. | 2020 | BC | restricted areas | enhanced HVAC system | unclear | HEPA | MAF |
| | | | BC | restricted areas | enhanced HVAC system | unclear | HEPA | TAF |
| | | | FC | restricted areas | enhanced HVAC system | unclear | HEPA | MAF |
| | | | FC | restricted areas | enhanced HVAC system | unclear | HEPA | TAF |
| [47] | Dougall et al. | 2019 | BC | restricted areas | enhanced HVAC system | unclear | HEPA | Unclear |
| [48] | Sossai et al. | 2011 | BC | restricted areas | conventional HVAC system | 12.5 air changes per hour | unclear | TAF |
| | | | BC | restricted areas | enhanced HVAC system | 12.5 air changes per hour | HEPA | additional LAF screen |
| [49] | Cristina et al. | 2016 | BC | restricted areas | enhanced HVAC system | 19 efficacious air exchanges per hour | HEPA | TAF |
| [18] | Andersson et al. | 2014 | BC | restricted areas | conventional HVAC system | unclear | F9 | displacement ventilation |
| | | | BC | restricted areas | enhanced HVAC system | unclear | HEPA | LAF |
| [50] | Perdelli, Cristina et al. | 2006 | FC | restricted areas | enhanced HVAC system | at least 15 air exchanges per hour | HEPA | unclear |
| | | | FC | inpatient facilities | conventional HVAC system | 6 air exchanges per hour | filters with 80%-85% efficiency | unclear |
| | | | FC | publicly accessible areas | conventional HVAC system | 2 air exchanges per hour | filters with 80%-85% efficiency | unclear |
| [51] | Sixt et al. | 2007 | FC | inpatient facilities | enhanced HVAC system | unclear | HEPA | LAF |
| | | | FC | inpatient facilities | conventional HVAC system | unclear | Plasmair™ | without LAF |
| | | | FC | inpatient facilities | without HVAC system | NA | NA | NA |

(*Continued*)

**Table 1.** (Continued)

| Reference | Author | year | Microbiological count type | Hospital locations tested | Ventilation method | ACH | Filter type | Type of ventilation system |
|---|---|---|---|---|---|---|---|---|
| [52] | Agodi et al. | 2015 | BC | restricted areas | enhanced HVAC system | 18 ± 4.5 air changes per hour | HEPA | LAF |
| [52] | Agodi et al. | 2015 | BC | restricted areas | enhanced HVAC system | 18 ± 4.5 air changes per hour | HEPA | MAF |
| | | | BC | restricted areas | enhanced HVAC system | 18 ± 4.5 air changes per hour | HEPA | TAF |
| [21] | Alsved et al. | 2018 | BC | restricted areas | enhanced HVAC system | unclear | HEPA | turbulent mixed air flow |
| | | | BC | restricted areas | enhanced HVAC system | unclear | HEPA | LAF |
| | | | BC | restricted areas | enhanced HVAC system | unclear | HEPA | temperature controlled air flow |
| [53] | Cho et al. | 2018 | FC | inpatient facilities | conventional HVAC system | unclear | without HEPA | unclear |
| | | | FC | inpatient facilities | enhanced HVAC system | unclear | HEPA | unclear |
| [54] | Curtis et al. | 2005 | FC | inpatient facilities | conventional HVAC system | 2.9 ± 2.7 air changes per hour | without HEPA | unclear |
| | | | FC | inpatient facilities | conventional HVAC system | 8.1 ± 7.9 air changes per hour | without HEPA | unclear |
| | | | FC | inpatient facilities | conventional HVAC system | 3.8 ± 4.0 air changes per hour | without HEPA | Unclear |
| [54] | Curtis et al. | 2005 | FC | inpatient facilities | enhanced HVAC system | 26.6 ± 13.2 air changes per hour | HEPA | unclear |
| [55] | Dehghani et al. | 2018 | FC | restricted areas | enhanced HVAC system | unclear | HEPA | LAF |
| [56] | Kabir et al. | 2012 | BC | inpatient facilities | conventional HVAC system | unclear | unclear | unclear |
| [57] | Falvey et al. | 2007 | FC | publicly accessible areas | conventional HVAC system | unclear | 65% filtering efficiency of fan | unclear |
| | | | FC | publicly accessible areas | conventional HVAC system | unclear | 65% filtering efficiency of fan | unclear |
| | | | FC | publicly accessible areas | conventional HVAC system | unclear | 90–95% filtering efficiency of fan | unclear |
| | | | FC | publicly accessible areas | conventional HVAC system | unclear | 90–95% filtering efficiency of fan | unclear |
| | | | FC | inpatient facilities | conventional HVAC system | 3 air exchanges per hour | 90–95% filtering efficiency of fan | unclear |
| | | | FC | inpatient facilities | conventional HVAC system | 3 air exchanges per hour | 90–95% filtering efficiency of fan | unclear |
| | | | FC | inpatient facilities | conventional HVAC system | 6 air exchanges per hour | 90–95% filtering efficiency of fan | unclear |
| | | | FC | inpatient facilities | conventional HVAC system | 6 air exchanges per hour | 90–95% filtering efficiency of fan | unclear |
| | | | FC | inpatient facilities | enhanced HVAC system | 12 air exchanges per hour | HEPA | Unclear |

(*Continued*)

**Table 1.** (Continued)

| Reference | Author | year | Microbiological count type | Hospital locations tested | Ventilation method | ACH | Filter type | Type of ventilation system |
|---|---|---|---|---|---|---|---|---|
| [57] | Falvey et al. | 2007 | FC | inpatient facilities | enhanced HVAC system | 12 air exchanges per hour | HEPA | Unclear |
| [58] | Friberg et al. | 2001 | BC | restricted areas | enhanced HVAC system | unclear | HEPA | LAF |

BC, bacterial count; FC, fungal count; NA, not applicable; HEPA, high efficiency particulate air; LAF, laminar air flow; TAF, turbulent air flow; MAF, mixed air flow.

m$^3$; fungal count: 5.46 ± 2.77 CFU/m$^3$) were significantly lower than those in TAF conditions (bacterial count: 36.13 ± 38.29 CFU/m$^3$; fungal count: 0.09 ± 0.07 CFU/m$^3$), while the results of fungal count were opposite.

The results of methodological quality evaluation are shown in Table 6, and the scores are between 3–7 points. All the included data were from survey samples, but no inclusion/exclusion criteria were established for the sampling conditions, and no blind method was adopted. We could not be sure if the researchers took follow-up samples from the study environment. In addition, 72.7% of the studies described the time when the survey was conducted, 63.6% repeated sampling, 36.4% explained the reason for excluding the samples, 9.1% described the treatment of missing data, and 27.3% calculated the positive rates of the samples.

The results of the meta-analysis showed that compared with the conventional HVAC systems used in restricted areas, the indoor bacterial count of LAF system conditions decreased by 40.05 CFU/m$^3$ (95%CI: -55.52, -24.58) (Table 6). Moreover, the use of a HEPA filter reduced the bacterial count by 113.14 CFU/m$^3$ (95%CI: -197.89, -28.38) (Table 7 and Fig 6) and the fungal count by 6.53 CFU/m$^3$ (95%CI: -10.50, -2.55) (Table 8 and Fig 7) compared with not using a HEPA filter. Further, in the sensitivity analysis, the conclusion that the LAF system and HEPA filter are effective remained stable. However, all of the results showed high heterogeneity. The results of the Begg's Test on publication bias indicated a p > 0.05 outcome. All the studies included in the meta-analysis were observational studies, and the results of the methodological quality assessment are shown in Table 9 and Fig 8.

## Conclusions

### Hospital areas comparisons

It is typically thought that the concentration of microbial bioaerosols should be lower in restricted areas of hospitals because of more stringent management and disinfection measures. However, according to our statistical results, in the hospitals that did not use HVAC systems and enhanced HVAC systems, the mean bacterial count in the restricted areas (36.12 CFU/m$^3$) was higher than that in the inpatient facilities (20 CFU/m$^3$) (Fig 4). This result may have been caused by our small sample size. Samples from the room with enhanced HVAC system in the inpatient facilities were obtained from only one article, with 42 bacterial samples in total, accounting for 1.02% of the total bacterial samples. Meanwhile, 34 bacterial samples from 2 papers were obtained from a room with a conventional HVAC system in a restricted area, accounting for 0.83% of the total bacterial samples. Nevertheless, this reminds us that the low bioaerosol concentrations in restricted areas should not be taken for granted. High air quality in restricted areas m requires higher investment compared to that in other areas.

In publicly accessible areas of the hospitals, no bacterial samples met our inclusion criteria, and all 575 fungal samples from these areas came from conditions wherein a conventional

**Table 2. Characteristics of Studies included in meta-analysis.**

| study | study period | country | sampling conditions | intervention | outcomes |
|---|---|---|---|---|---|
| Hansen et al. (2005) | unclear | Germany | 105 septic/aseptic operation procedures in restricted areas | G1: laminar air flow system with HEPA; | bacterial count |
| | | | | G2: conventional HVAC system without LAF and HEPA | |
| Andersson et al. (2014) | April 2010—May 2011 | Sweden | 63 orthopedic implant operations in restricted areas | G1: laminar air flow system with HEPA; | bacterial count |
| | | | | G2: displacement ventilation system without HEPA | |
| Agodi et al. (2015) | March 2010—February 2011 | Italy | 1228 elective prosthesis procedures (60.1% hip and 39.9% knee) in restricted areas | G1: laminar air flow system with HEPA; | bacterial count |
| | | | | G2: turbulent air flow or mixed air flow system with HEPA | |
| Alsved et al. (2018) | January 2015—February 2016 | Sweden | 45 operations (21 wrist fractures, 6 shoulder arthroscopies, and 18 hip fracture fixations) in restricted areas | G1: laminar air flow system with HEPA; | bacterial count |
| | | | | G2: turbulent mixed air flow system with HEPA | |
| Perdelli, Sartini et al. (2006) | unclear | Genova and Rome, Italy | no operation performed | G1: turbulent air flow system with HEPA; | bacterial count, fungal count |
| | | | | G2: turbulent air flow system with filters with 80% to 85% efficiency | |
| Perdelli, Cristina et al. (2006) | unclear | Italy | unclear | G1: HVAC system with HEPA; | fungal count |
| | | | | G2: conventional HVAC system with filters with 80% to 85% efficiency | |
| Brun et al. (2013) | December 2009—January 2011 | Brazil | no operation performed | G1: HVAC system with HEPA; | fungal count |
| | | | | G2: conventional HVAC system without HEPA | |
| Curtis et al. (2005) | September 1998—September 1999 | United States | no operation performed | G1: HVAC system with HEPA; | fungal count |
| | | | | G2: conventional HVAC system without HEPA | |
| Falvey et al. (2007) | 1995–2005 | United States | no operation performed | G1: HVAC system with HEPA; | fungal count |
| | | | | G2: conventional HVAC system with filters with 95% efficiency | |
| Sixt et al. (2007) | December 2004—January 2006 | France | unclear | G1: laminar air flow system with HEPA; | fungal count |
| | | | | G2: conventional HVAC system without HEPA | |
| Cho et al. (2018) | May 2017—May 2018 | South Korea | unclear | G1: HVAC system with HEPA; | fungal count |
| | | | | G2: conventional HVAC system without HEPA | |

G1, the intervention group; G2, the control group; SD, standard deviation; LAF, laminar air flow; HEPA, high-efficiency particulate air.

HVAC system. We conducted meta-analysis for samples from inpatient facilities and restricted areas because most of the bacterial samples came from restricted areas (n = 3739, 91.20%) (Table 3 and Fig 2) and most of the fungal samples came from inpatient facilities (n = 4015, 76.68%) (Table 4 and Fig 3).

## Ventilation comparisons

The mean counts of bacteria in the conditions without HVAC systems, with conventional HVAC systems, and with enhanced HVAC systems decreased in turn in the inpatient facilities

**Table 3. Mean bacterial counts in different areas of hospitals (CFU/m³).**

| | inpatient facilities | | | | restricted areas | | | |
|---|---|---|---|---|---|---|---|---|
| | without HVAC | conventional HVAC | enhanced HVAC | total | without HVAC | conventional HVAC | enhanced HVAC | total |
| n(studies) | 61(2) | 258(4) | 42(1) | 361(4) | 10(1) | 34(2) | 3695(16) | 3739(16) |
| mean | 356.45 | 229.24 | 20 | 226.39 | 387.50 | 130.01 | 36.12 | 37.91 |
| SD | 177.18 | 106.02 | NA | 145.25 | NA | 153.55 | 39.74 | 46.64 |
| 95%CI | 311.07–401.82 | 216.24–242.24 | NA | 211.36–241.43 | NA | 76.44–183.59 | 34.84–37.40 | 36.42–39.41 |
| median | 265 | 151.49 | 20 | 151.49 | 387.50 | 25.40 | 16.63 | 16.63 |
| range | 265–694 | 130–407 | 20 | 20–694 | 387.50 | 25–349 | 0–279 | 0–388 |

HVAC, heating, ventilation and air conditioning; n, number of samples; SD, standard deviation; CI, confidence interval; NA, not applicable.

and restricted areas of the hospitals (p<0.05) (Table 3 and Fig 4). The results for fungi were the same as those for bacteria, but there was a lack of data for publicly accessible and restricted areas (Table 4 and Fig 5). These results indicated that conventional HVAC systems effectively removed bacteria and fungi from indoor air, and that enhanced HVAC systems were more effective than conventional HVAC systems.

Enhanced HVAC systems use HEPA filters or LAF systems, or have higher ACH than conventional HVAC systems. According to our statistical results (Table 5), the HEPA filters used in enhanced HVAC systems proved effective in reducing both bacterial and fungal concentrations in the room. High ACH effectively reduced the indoor fungal concentration, but there was no significant difference in the ability to remove bacteria (p = 0.175). For LAF systems, bacterial and fungal counts showed opposite results, with lower bacterial counts and higher fungal counts in the air in LAF rooms than in TAF rooms. All operating rooms equipped with laminar flow systems were also equipped with HEPA filters, so the protective effect of unidirectional air flow in LAF systems still needs to be further analysed (Table 5).

## HEPA filters

In the subgroup analysis of fungal CFU concentrations for HEPA filter conditions, we found that the incubation temperatures after sampling significantly affected the results. This is because colonies may not grow properly at uncomfortable incubation temperatures, making it impossible to calculate an accurate microbial concentration. For example, when fungi were incubated at 25°C, the CFU concentration of fungi in the air—which were converted according to the incubation results—in the room with a HEPA filter was lower than that in the room without such a filter (22.15 CFU/m3; 95%CI: -35.79, -8.50). This gap was reduced to 3.32

**Table 4. Mean fungal counts in different areas of hospitals (CFU/m³).**

| | publicly accessible areas | inpatient facilities | | | | | restricted areas |
|---|---|---|---|---|---|---|---|
| | conventional HVAC | natural ventilation | without HVAC | conventional HVAC | enhanced HVAC | total | enhanced HVAC |
| n(studies) | 575(3) | 32(1) | 140(3) | 3046(10) | 797(6) | 4015(10) | 646(5) |
| mean | 61.21 | 1000.00 | 38.17 | 32.82 | 13.67 | 36.91 | 4.27 |
| SD | 59.35 | 33.02 | 101.36 | 85.75 | 10.98 | 116.11 | 3.13 |
| 95%CI | 56.35–66.08 | 988.10–1011.90 | 21.24–55.11 | 29.77–35.86 | 12.9–14.43 | 33.32–40.50 | 4.03–4.51 |
| median | 65 | 1000 | 9.16 | 9.19 | 9.73 | 9.73 | 5.28 |
| range | 11–245 | 968–1033 | 1–354 | 0–710 | 0–41 | 0–1033 | 0–8 |

HVAC, heating, ventilation and air conditioning; n, number of samples; SD, standard deviation; CI, confidence interval; NA, not applicable.

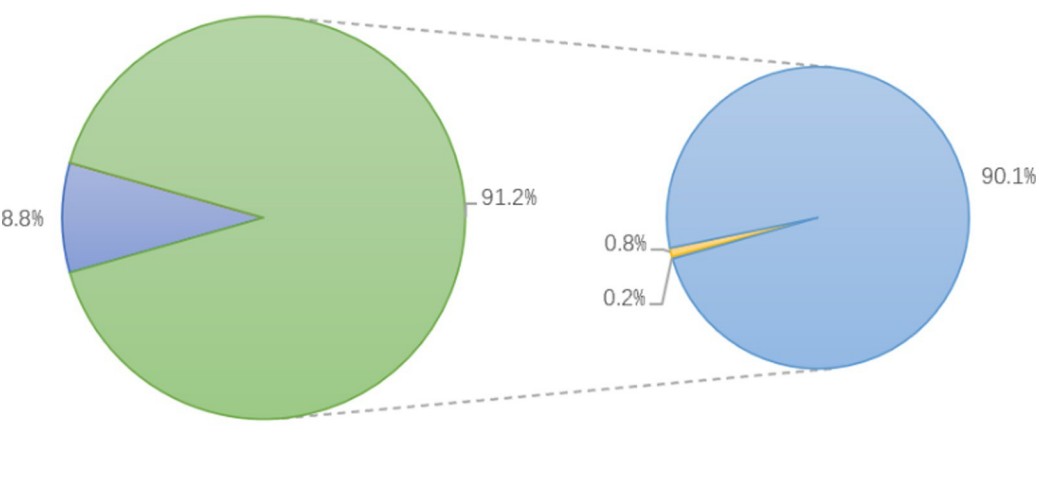

inpatient facilities ■restricted areas ■without HVAC ■conventional HVAC ■enhanced HVAC

**Fig 2. The distribution of bacterial samples from different areas.**

CFU/m$^3$ (95%CI: -4.60, -2.04) at incubation temperatures of 30˚C. If the incubation temperature was increased to 35˚C–37˚C, the resulting effect range was found to cross the no-effect line (95%CI: -6.52, 0.24) (Table 8 and Fig 8).

Based on the above results, we can conclude that the HEPA filter is effective in reducing the concentration of fungi in hospital indoor air, and its effectiveness can be demonstrated at the appropriate incubation temperature. For bacterial results, the difference in incubation temperature did not seem to be the main cause of excessive heterogeneity (Table 7). Rather, it may be attributed to other condition factors such as the HVAC system's cleaning frequency, filter's replacement cycle, number of people in the room, frequency of door opening, additional disinfection regimens, the type of culture medium used in microbial counting, and so on. The source of heterogeneity was not found due to the insufficient number of included articles.

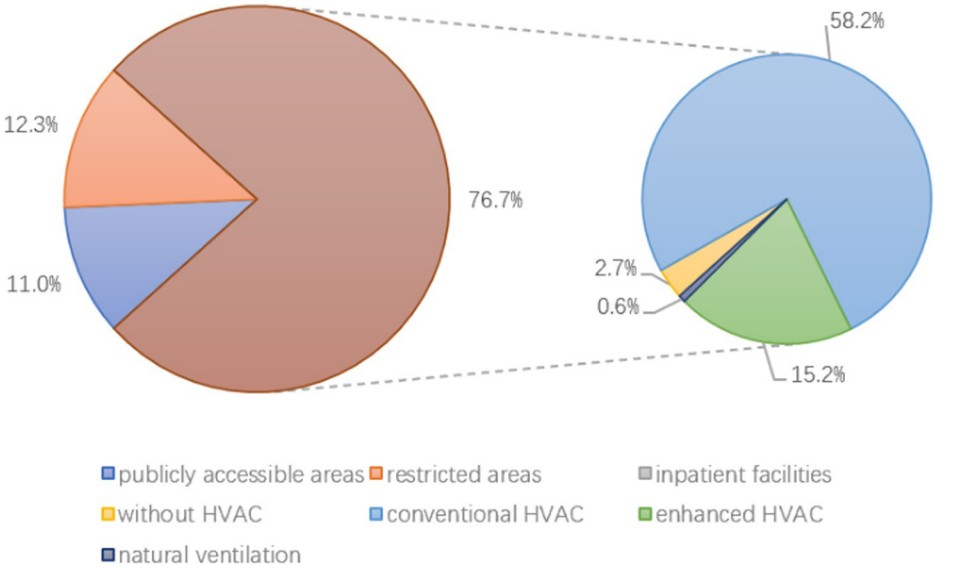

■publicly accessible areas ■restricted areas        ■inpatient facilities
■without HVAC        ■conventional HVAC        ■enhanced HVAC
■natural ventilation

**Fig 3. The distribution of fungal samples from different areas.**

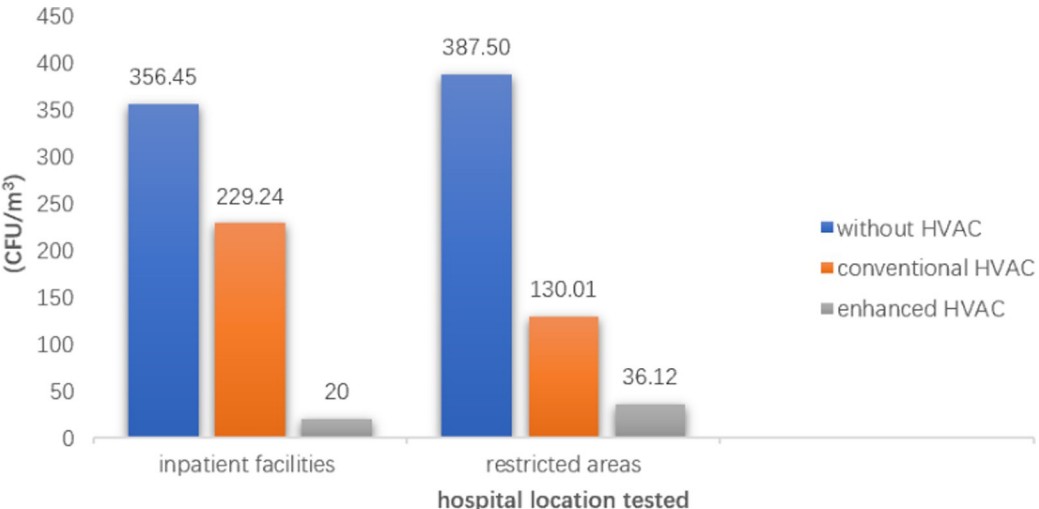

**Fig 4. Mean bacterial colony forming units per cubic meter (CFU/m³) in hospitals.**

Overall, the use of HEPA filters reduced bacteria concentration by 113.14 CFU/m³ (95%CI: -197.89, -28.38) in hospital indoor air.

## LAF systems

All the studies in our meta-analysis [21, 33, 52, 59] conducted sampling during operation procedures, while the study by Agodi et al. [52] also included samples taken during the non-working state. These four studies provide information on the differences in the sample areas of operating rooms. Samples from Hansen et al. [33], Andersson et al. [59], and Agodi et al. [52]. collected samples from places as near as possible to the operating area (maximum distance 50 cm), in the surgical wound area, and close to the wound, respectively. Alsved et al. [21] collected samples from 1 m above the operating table, at the instrument table, and in the periphery of the room. In addition, we excluded the temperature controlled air flow system from our meta-analysis because it is a newly developed ventilation system [21].

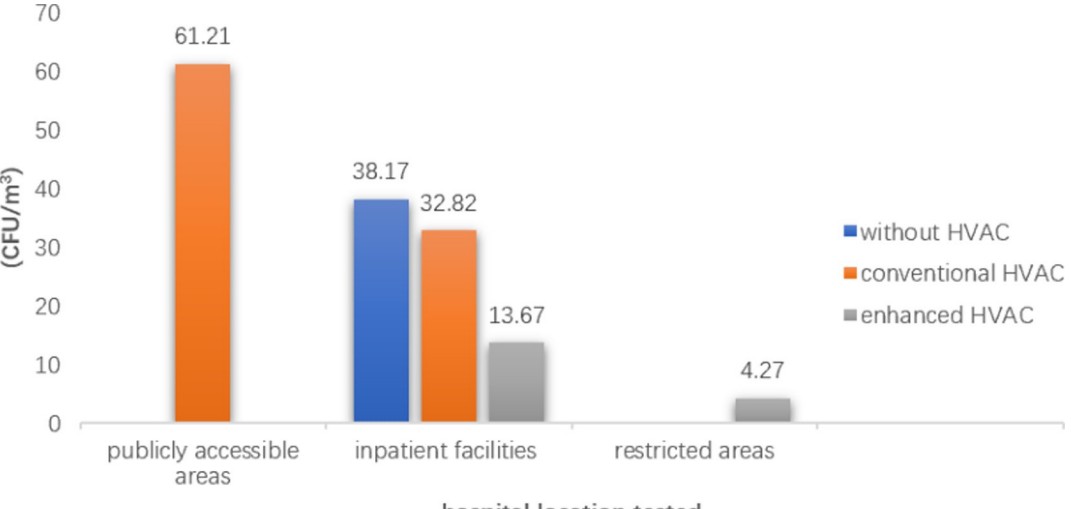

**Fig 5. Mean fungal colony forming units per cubic meter (CFU/m³) in hospitals.**

**Table 5. Mean colony forming units per cubic meter (CFU/m$^3$) sampled from all of the areas.**

| | Bacterial counts | | | | | | Fungal counts | | | | | |
|---|---|---|---|---|---|---|---|---|---|---|---|---|
| | n(studies) | mean | SD | 95% CI | median | range | n(studies) | mean | SD | 95% CI | median | range |
| **Type of HVAC** | | | | | | | | | | | | |
| LAF | 1651(6) | 26.28 | 29.78 | 24.85–27.72 | 6.86 | 3–78 | 299(2) | 5.46 | 2.77 | 5.32–5.95 | 7.89 | 2–8 |
| TAF | 819(9) | 36.13 | 38.29 | 33.51–38.76 | 12.90 | 1–130 | 126(2) | 0.09 | 0.07 | 0.08–0.10 | 0.10 | 0–0.16 |
| **ACH** | | | | | | | | | | | | |
| ≥ 15 exchanges per hour | 1592(9) | 53.45* | 47.15 | 51.13–55.77 | 35 | 2–279 | 317(3) | 12.21 | 14.22 | 10.64–13.78 | 5.28 | 5–41 |
| < 15 exchanges per hour | 136(2) | 58.05* | 53.53 | 48.98–67.13 | 25.4 | 10–130 | 2899(4) | 22.96 | 17.17 | 22.33–23.58 | 14.26 | 0–84 |
| **Filter** | | | | | | | | | | | | |
| HEPA filters | 3590(16) | 36.90 | 40.06 | 35.59–38.21 | 16.63 | 0–279 | 1443(11) | 9.46 | 9.63 | 8.96–9.96 | 7.89 | 0–41 |
| Other filters | 206(3) | 57.83 | 85.06 | 46.15–69.52 | 12.50 | 13–349 | 3555(8) | 27.85 | 33.41 | 26.75–28.95 | 14.26 | 0–245 |

HVAC, heating, ventilation and air conditioning; LAF, laminar air flow; TAF, turbulent air flow; ACH, air change per hour; HEPA, high-efficiency particulate air; n, number of samples; SD, standard deviation; CI, confidence interval; *, There was no statistical difference between the two groups (p = 0.175).

Table 6 shows the bacterial concentrations measured in LAF and non-LAF conditions. Bacterial CFU concentration was 40.05 CFU/m$^3$ (95%CI: -55.52, -24.58) lower in indoor

**Table 6. Methodological quality assessment.**

| | Item 1: Source of Information | Item 2: Inclusion/ Exclusion Criteria | Item 3: Time Period for Identity | Item 4: Subjects consecutive | Item 5: Evaluat-ors Masked | Item 6: Quality Assurance Assessments | Item 7: Samples Exclusions | Item 8: Confoundi-ng assessed/ controlled | Item 9: Missing Data | Item 10: Response Rates | Item 11: Follow-up | Total Items Reported (Max. 11) |
|---|---|---|---|---|---|---|---|---|---|---|---|---|
| Hansen et al. (2005) | Yes | Unclear | No | Yes | No | Yes | Yes | Yes | Unclear | No | Unclear | 5 |
| Andersson et al. (2014) | Yes | Unclear | Yes | Yes | No | Yes | Yes | Yes | Unclear | No | Unclear | 6 |
| Agodi et al. (2015) | Yes | No | Yes | Yes | No | Yes | No | Yes | No | No | Unclear | 5 |
| Alsved et al. (2018) | Yes | No | Yes | Yes | No | Yes | No | Yes | No | No | Unclear | 5 |
| Perdelli, Sartini et al. (2006) | Yes | No | No | Yes | No | Yes | No | Unclear | No | No | Unclear | 3 |
| Perdelli, Cristina et al. (2006) | Yes | No | No | Yes | No | No | No | Yes | No | No | Unclear | 3 |
| Brun et al. (2013) | Yes | Unclear | Yes | Yes | No | Unclear | No | Unclear | No | No | Unclear | 3 |
| Curtis et al. (2005) | Yes | Unclear | Yes | Yes | No | Yes | Unclear | Unclear | No | Yes | Unclear | 5 |
| Falvey et al. (2007) | Yes | No | Yes | Yes | No | Unclear | Unclear | Unclear | No | Yes | Unclear | 4 |
| Sixt et al. (2007) | Yes | No | Yes | Yes | No | No | Yes | Yes | Yes | Yes | Unclear | 7 |
| Cho et al. (2018) | Yes | No | Yes | Yes | No | Yes | Yes | Yes | No | No | Unclear | 6 |
| % Items (+) reported: | 100% | 0% | 72.7% | 100% | 0% | 63.6% | 36.4% | 63.6% | 9.1% | 27.3% | 0% | |

**Table 7. Meta-analysis comparing the mean bacterial colony forming units per cubic meter (CFU/m³) in OTs with LAF vs OTs without LAF.**

| | LAF | | | without LAF | | | weight | mean difference (95% CI) |
|---|---|---|---|---|---|---|---|---|
| | mean | SD | n | mean | SD | n | | |
| incubated at 30˚C | | | | | | | | |
| Andersson et al. (2014) | 1.00 | 2.10 | 164 | 15.90 | 13.40 | 91 | 32.8% | -14.90 [-17.67, -12.13] |
| *subtotal (95% CI)* | | | *164* | | | *91* | *32.8%* | *-14.90 [-17.67, -12.13]* |
| incubated at 35˚C-37˚C | | | | | | | | |
| Agodi et al. (2015) (1) | 22.08 | 34.61 | 126 | 279.42 | 128.73 | 21 | 6.3% | -257.34 [-312.73, -201.95] |
| Agodi et al. (2015) (2) | 22.08 | 34.61 | 126 | 62.23 | 45.02 | 62 | 27.1% | -40.15 [-52.88, -27.42] |
| Alsved et al. (2018) | 3.01 | 6.12 | 272 | 16.63 | 20.17 | 235 | 32.8% | -13.62 [-16.30, -10.94] |
| Hansen et al. (2005) | 6.86 | 30.32 | 652 | 348.75 | 251.27 | 11 | 1.1% | -341.89 [-490.40, -193.38] |
| *subtotal (95% CI)* | | | *1176* | | | *329* | *67.2%* | *-112.33[-165.32, -59.34]* |
| Total | | | 1340 | | | 420 | 100% | -40.05 [-55.52, -24.58] |

Test for heterogeneity showed very high inconsistency between the studies (I² = 96%).

Test for publication bias p = 0.086.

OT, operating theatres; LAF, laminar air flow; SD, standard deviation; n, number of samples; CI, confidence interval.

conditions with LAF system than in indoor conditions without LAF system. In our included study, all operating rooms equipped with LAF systems were equipped with HEPA filters. The HEPA filters reduced bacteria by 113.14 CFU/m³ (95%CI: -197.89, -28.38) compared to other filters (Table 7). Therefore, the LAF system has a negative effect on reducing the concentration

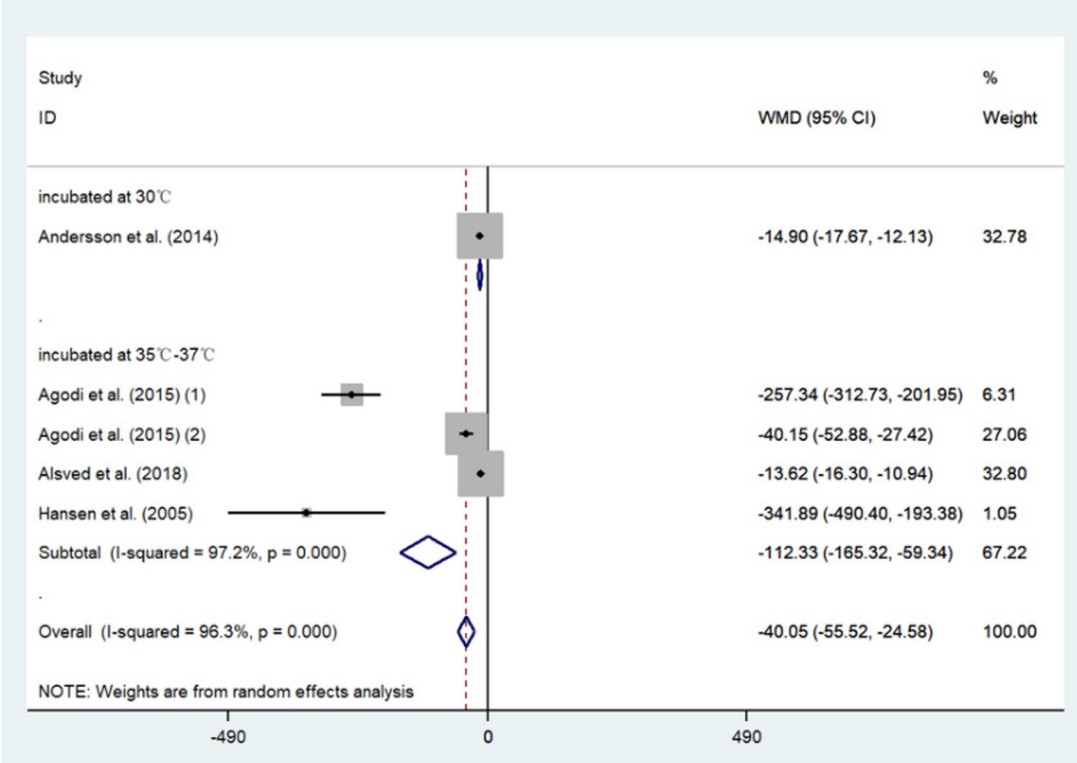

**Fig 6. Forest plots of comparing the mean bacterial colony forming units per cubic meter (CFU/m³) in OTs with LAF vs OTs without LAF.**

**Table 8. Meta-analysis comparing the mean bacterial colony forming units per cubic meter (CFU/m³) in rooms with HEPA vs rooms without HEPA.**

| | HEPA | | | without HEPA | | | weight | mean difference (95% CI) |
|---|---|---|---|---|---|---|---|---|
| | mean | SD | n | mean | SD | n | | |
| incubated at 30˚C | | | | | | | | |
| Andersson et al. (2014) | 1.00 | 2.10 | 164 | 15.90 | 13.40 | 91 | 40.9% | -14.90 [-17.67, -12.13] |
| *subtotal (95% CI)* | | | *164* | | | *91* | *40.9%* | *-14.90 [-17.67, -12.13]* |
| incubated at 37˚C | | | | | | | | |
| Hansen et al. (2005) | 6.86 | 30.32 | 652 | 348.75 | 251.27 | 11 | 18.1% | -341.89 [-490.40, -193.38] |
| Perdelli, Sartini et al. (2006) | 20.00 | 6.42 | 42 | 130.00 | 13.78 | 48 | 40.9% | -110.00 [-114.35, -105.65] |
| *subtotal (95% CI)* | | | *694* | | | *59* | *59.1%* | *-213.58[-439.53,12.38]* |
| Total | | | 858 | | | 150 | 100% | -113.14 [-197.89, -28.38] |

Test for heterogeneity showed very high inconsistency between the studies ($I^2$ = 99.8%).

Test for publication bias p = 1.

OT, operating theatres; LAF, laminar air flow; SD, standard deviation; n, number of samples; CI, confidence interval.

of bacteria in indoor air, and the use of LAF system instead weakened the HEPA filters' effect. We hypothesized that the presence of only HEPA filters and conventional HVAC systems in the operating room might have a higher bacterial removal effect.

As for the influence of LAF systems on the CFU concentration of fungi in the air, we did not analyse the systems' effectiveness of the LAF system based on fungi count because too few fungal samples met our inclusion criteria. However, according to the statistical description of the results, a higher concentration of fungi was observed in the operating room with a LAF

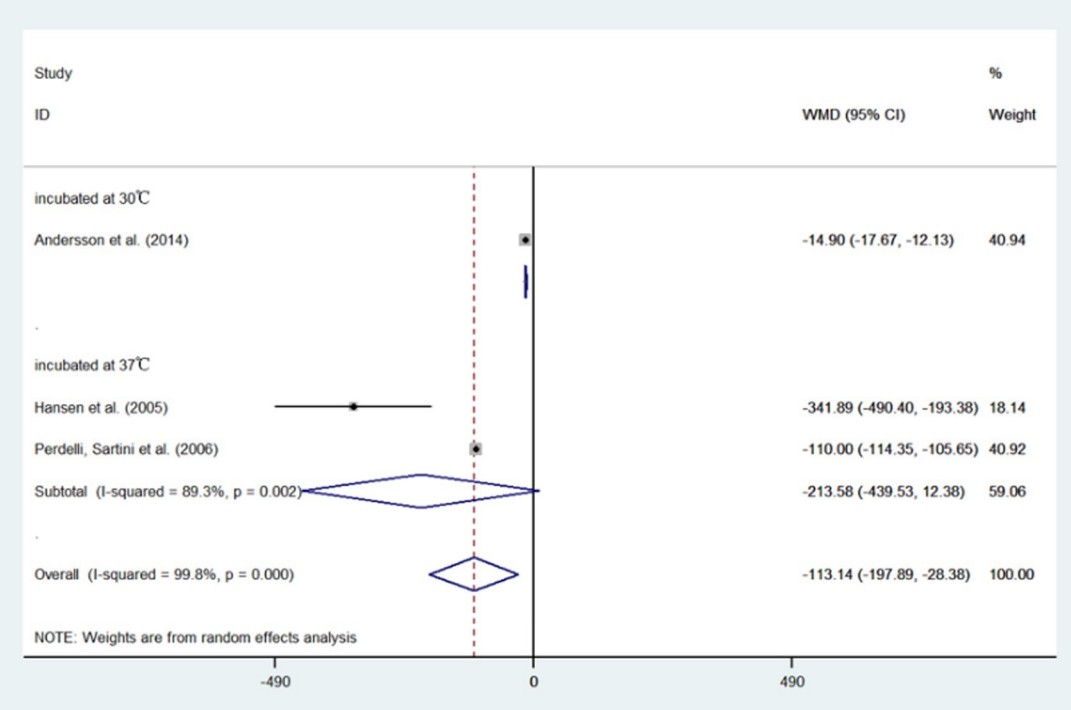

**Fig 7. Forest plots of comparing the mean bacterial colony forming units per cubic meter (CFU/m³) in rooms with HEPA vs rooms without HEPA.**

**Table 9. Meta-analysis comparing the mean fungal colony forming units per cubic meter (CFU/m$^3$) in rooms with HEPA vs rooms without HEPA.**

| | HEPA | | | without HEPA | | | weight | mean difference (95% CI) |
|---|---|---|---|---|---|---|---|---|
| | mean | SD | n | mean | SD | n | | |
| incubated at 25˚C | | | | | | | | |
| Brun et al. (2013) | 25.20 | 20.40 | 26 | 110.30 | 78.10 | 26 | 1.5% | -85.10 [-116.13, -54.07] |
| Curtis et al. (2005) | 41.00 | 65.00 | 62 | 83.50 | 113.00 | 71 | 1.6% | -42.50 [-73.36, -11.64] |
| Falvey et al. (2007) 25˚C | 20.04 | 29.56 | 249 | 26.00 | 21.00 | 93 | 16.6% | -5.96 [-11.59, -0.33] |
| Perdelli, Cristina et al. (2006) | 5.00 | 11.00 | 65 | 14.26 | 10.02 | 310 | 22.0% | -9.26 [-12.16, -6.36] |
| *subtotal (95% CI)* | | | *402* | | | *500* | *41.7%* | *-22.15 [-35.79, -8.50]* |
| incubated at 30˚C | | | | | | | | |
| Sixt et al. (2007) | 2.24 | 0.98 | 119 | 5.56 | 10.15 | 245 | 24.3% | -3.32 [-4.60, -2.04] |
| *subtotal (95% CI)* | | | *119* | | | *245* | *24.3%* | *-3.32 [-4.60, -2.04]* |
| incubated at 35˚C-37˚C | | | | | | | | |
| Cho et al. (2018) | 0.35 | 2.01 | 50 | 4.10 | 4.18 | 25 | 23.8% | -3.75 [-5.48, -2.02] |
| Falvey et al. (2007) 37˚C | 9.73 | 70.21 | 249 | 8.00 | 20.00 | 93 | 10.1% | 1.73 [-7.89, 11.35] |
| *subtotal (95% CI)* | | | *299* | | | *118* | 34.0% | *-3.14 [-6.52, 0.24]* |
| Total | | | 820 | | | 863 | 100% | -6.53 [-10.50, -2.55] |

Test for heterogeneity showed very high inconsistency between the studies ($I^2$ = 87.4%).

Test for publication bias p = 0.133.

OT, operating theatres; LAF, laminar air flow; SD, standard deviation; n, number of samples; CI, confidence interval.

system (5.46 ± 2.77 CFU/m$^3$) compared with the operating room with a TAF system (0.09 ± 0.07 CFU/m$^3$) (Table 5). The effectiveness of HEPA filters in removing fungi from the air has been determined (Table 8). Therefore, we ultimately conclude that, other components of the LAF system weakened the HEPA filter's ability to remove bacteria and fungi.

## Discussion

We investigated airborne concentrations of bacterial and fungal CFU in various areas of the hospital environment and looked for correlations with HVAC systems. We found that the use of HVAC systems could effectively remove these bacteria and fungi. Moreover, the use of HEPA filters in inpatient facilities and restricted areas reduced bacteria by an average of 113.14 CFU/m$^3$ and fungi by 6.53 CFU/m$^3$. The above results fluctuated according to the different incubation temperatures after sampling, especially for the cultivation of fungi, where the temperature may have a great influence on the final converted CFU concentrations.

In the existing LAF system, other parts other than the HEPA filters did not seem to work as they should. This is because according to our statistical results, the use of LAF systems in the operating room only reduced bacteria by 40.05 CFU/m$^3$, less than the CFU reduction of HEPA filters. That is to say, HEPA filters really play a vital role in removing bioaerosols in operating rooms, while other LAF system designs may not be as efficient compared with TAF system. To some extent, this result explains the ineffectiveness of LAF systems in reducing surgical site infections (SSIs).

The study by Hansen et al. reported no differences in bioaerosol concentrations during operation procedures by operation type, number of participating people, and the material of the clothes [33]. Further, in the study by Andersson et al., the number of people present in the operating room the door opening frequency affected bioaerosol concentration significantly, especially in displacement ventilation operating rooms [18]. Additionally, the study by Agodi et al. confirmed that the frequency of door opening and the number of people in an operation

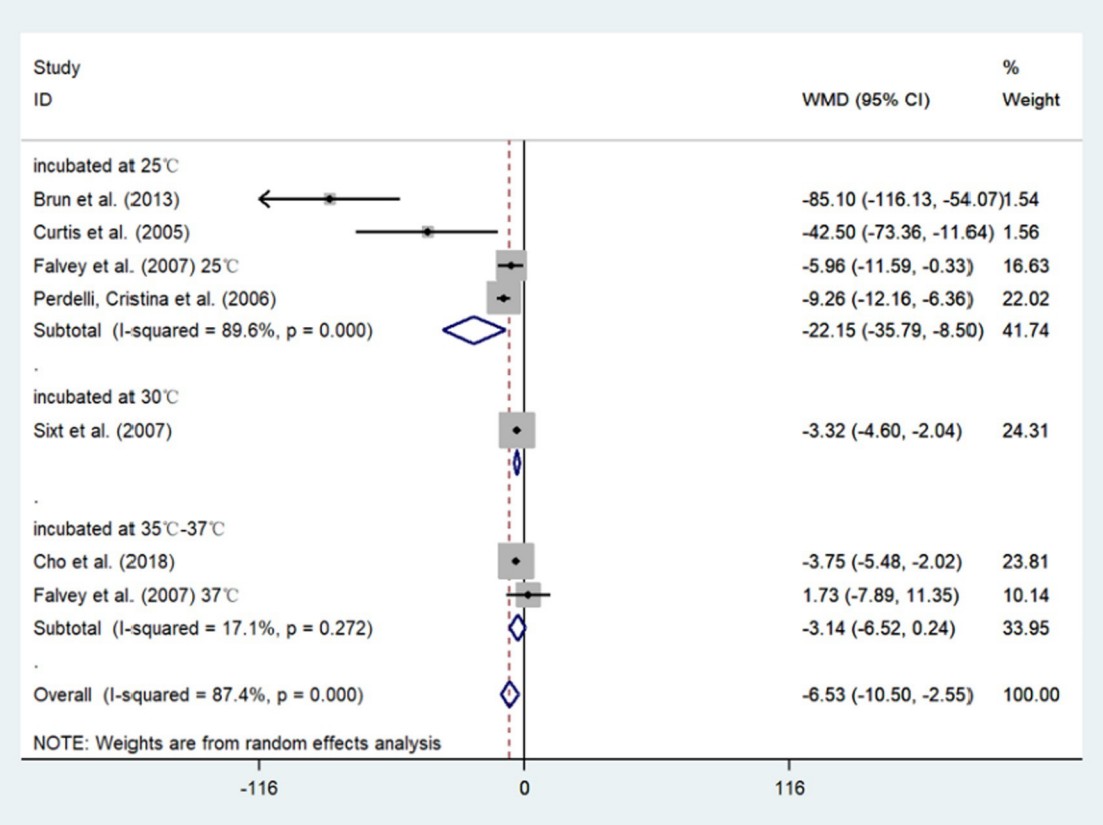

**Fig 8. Forest plots of comparing the mean fungal colony forming units per cubic meter (CFU/m3) in rooms with HEPA vs rooms without HEPA.**

room might be key factors in increasing bacterial counts [52]. Finally, Alsved et al. found neither the frequency of door openings nor people present during surgery to be correlated with bioaerosol concentrations [21]. In general, there were diametrically opposite conclusions about the number of people in a room and the frequency of opening doors. We supposed that an operating room can be viewed as a complex system with interactions between patients, different professional teams, and highly specialized techniques, and that it is characterized by the fact that small mistakes or failures can lead to serious adverse events [41]. Because this complex system involves numerous transient phenomena, the air flow distribution of the LAF systems was easily disturbed [39, 40]. In addition, there may be some problems with the current LAF systems. These may include inadequate plenum/canopy size due to undersized areas of ceiling-producing LAF, incorrect positioning of the instruments table (which needs to be entirely under the LAF canopy), and variable cooling of the operating room air—causing local wound area hypothermia and giving surgeons a false sense of sterility security—leading to unnoticed wound contamination during operating procedures [11].

This study possesses the following limitations. (1) Since all of the included studies were observational, it is likely that there are other factors that have not been analysed, such as a HVAC system's cleaning frequency, a filter's replacement cycle, number of people in the room, frequency of door opening, additional disinfection regimens, the type of culture medium used in microbial counting, and so on. (2) Incubation temperature had a great influence on the results of fungi. However, due to the insufficient number of included studies, we

did not classify and analyse these CFU concentrations at different incubation temperatures. Further, the effect of incubation temperature on bacterial outcomes was not reflected in the subgroup analysis, which may be due to the interference of the previously mentioned unanalysed influencing factors. Therefore, the bacterial/fungal removal amount included the results at all incubation temperatures. With sufficient stratified calculation of the results at each incubation temperature would be more appropriate. (3) Since we did not study specific microbial species and potential influences of chemical pollution [60], the overall colony count may be less meaningful. Especially concerning infectious diseases, it may be more beneficial to study specific microbes or viruses instead. Furthermore, even if the total concentrations of microbial cultures in the bioaerosols were similar, different inhalation risks are attributable to the different size distributions and compositions of bioaerosol particles [42]. (4) The results of the methodological quality assessment show that, of the 11 studies included in the meta-analysis, 63.6% of the studies did not explain the situation of discarded samples, 72.7% did not give a positive rate of samples, and 90.9% did not explain the treatment method of missing data. We only extracted the sample size according to the original text and did not make any adjustments. Therefore, there may be some deviation in the statistics of the sample size.

Studies concerning SARS-CoV-2 have shown that infection risks associated with using HVAC systems did not increase during the COVID-19 pandemic [61]. The study by Gola et al. on indoor air chemical pollution showed that a HVAC system was beneficial to improve indoor air quality [62]. Our results regarding bioaerosols showed that the HVAC systems in hospitals today could effectively reduce the indoor concentrations of bioaerosols. This gave us confidence to use air conditioning normally during the COVID-19 pandemic. The use of HEPA filters is an effective option for areas that are under-ventilated and require additional protection. However, the LAF system was not satisfactory in its ability to remove bioaerosols. Other components of the LAF system other than the HEPA filter were not conducive to removing airborne bacteria and fungi. It is important to note that choosing the best between IAQ and energy efficiency was not an easy task [60], and the routine maintenance and cleaning costs of HVAC systems were often not cheap, especially in indoor conditions with LAF systems. For example, HEPA filters must be replaced regularly because their filter materials can have variable or unknown gas adsorption and particle capture after long-term usage, which can cause a strong matrix effect [63]. Thus, both the purchase and maintenance costs of these enhanced HVAC systems should be taken into account [64]. Therefore, when deciding whether to use HEPA filters or LAF systems, specific cost-benefit analysis should be considered during the actual application process.

## Supporting information

**S1 File. Search query.**
(DOCX)

**S1 Appendix. Quality assessment forms.**
(DOCX)

**S1 Table. The formulas used for conversion.**
(DOCX)

**S2 Table. The data extracted from 35 studies included.**
(XLSX)

**S1 Checklist. Quality assessment forms.**
(DOCX)

## Acknowledgments

The authors would like to express our gratitude to all researchers who provided data for our systematic review. We also thank personnel in the hospital-acquired infection control department of the First Affiliated Hospital of Zhejiang Chinese Medical University for providing useful advice on laminar air flow operating rooms management.

## Author Contributions

**Conceptualization:** Rongchen Dai, Shan Liu, Qiushuang Li.

**Data curation:** Rongchen Dai, Shan Liu, Qiushuang Li, Hanting Wu, Li Wu.

**Formal analysis:** Shan Liu, Li Wu.

**Funding acquisition:** Conghua Ji.

**Investigation:** Hanting Wu.

**Methodology:** Rongchen Dai, Shan Liu, Qiushuang Li, Conghua Ji.

**Project administration:** Conghua Ji.

**Software:** Hanting Wu, Li Wu.

**Writing – original draft:** Rongchen Dai, Shan Liu, Qiushuang Li, Conghua Ji.

**Writing – review & editing:** Conghua Ji.

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
