## [Decision Letter · Decision Letter 0]

8 Oct 2021

PONE-D-21-19989

A systematic review and meta-analysis of indoor bioaerosols in hospitals: the influence of heating, ventilation and air conditioning

PLOS ONE

Dear Dr. Ji,

Thank you for submitting your manuscript to PLOS ONE. After careful consideration, we feel that it has merit but does not fully meet PLOS ONE’s publication criteria as it currently stands. Therefore, we invite you to submit a revised version of the manuscript that addresses the points raised during the review process.

We look forward to receiving your revised manuscript.

Kind regards,

Muhammad Shahzad Aslam, Ph.D.,M.Phil., Pharm-D

Academic Editor

PLOS ONE

2. Thank you for stating the following in the Acknowledgments/ funding Section of your manuscript:

“This work was supported by the key R & D projects from the Department of Science and Technology of Zhejiang Province [NO.2020C03126] and the Administration of Traditional Chinese Medicine of Zhejiang Province [NO.2017ZZ007], the People’s Republic of China.”

“This work was supported by the key R & D projects from the Department of Science and Technology of Zhejiang Province [NO.2020C03126] and the Administration of Traditional Chinese Medicine of Zhejiang Province [NO.2017ZZ007], the People’s Republic of China.”

Additional Editor Comments (if provided):

1-Please write down limitation of study as separate heading

2-Please include strength of the study as separate heading

3-What are the practical implication? Please write down in details

Reviewers' comments:

Reviewer's Responses to Questions

**Comments to the Author**

1. Is the manuscript technically sound, and do the data support the conclusions?

Reviewer #1: Yes

Reviewer #2: Yes

2. Has the statistical analysis been performed appropriately and rigorously? 

Reviewer #1: Yes

Reviewer #2: Yes

3. Have the authors made all data underlying the findings in their manuscript fully available?

Reviewer #1: Yes

Reviewer #2: Yes

4. Is the manuscript presented in an intelligible fashion and written in standard English?

Reviewer #1: Yes

Reviewer #2: No

5. Review Comments to the Author

Reviewer #1: I suggest you include that HVACs are for maintaining good indoor air quality. There is often no real ventilation strategy dedicated to usersand the activities carried out. Ventilation must be part of an organic strategy and measures

In the discussion read and quote the articles.

Gola m. capolongo S. et al.

Indoor air in healing environments: Monitoring chemical pollution in inpatient rooms. Facilities, 2019, 37(9-10), pp. 600–623.

Existing guidelines for indoor air quality: The case study of hospital environments

SpringerBriefs in Public Health, 2017, (9783319491592), pp. 13–26

Indoor Air Quality in Inpatient Environments: A Systematic Review on Factors that Influence Chemical Pollution in Inpatient Wards

Journal of Healthcare Engineering, 2019, 2019, 8358306.

Editorial

The Dichotomy between Indoor Air Quality and Energy Efficiency in Light of the Onset of the COVID-19 Pandemic

Atmosphere 2021, 12, 791. https://doi.org/10.3390/atmos12060791

Reviewer #2: Dear. Authors:

Some paragraphs were founded on critical grammar errors. Therefore, need to proofread one native researcher and provide the proofreading certificate supplementary file, and sent the revised version for peer review.

6. PLOS authors have the option to publish the peer review history of their article (what does this mean?). If published, this will include your full peer review and any attached files.

Reviewer #1: No

Reviewer #2: **Yes: **Dr. Kim Yun Jin, Ph.D

---

## [Author Response · Author response to Decision Letter 0]

24 Oct 2021

Response to Reviewer #1

1. Comment: I suggest you include that HVACs are for maintaining good indoor air quality. There is often no real ventilation strategy dedicated to usersand the activities carried out. Ventilation must be part of an organic strategy and measures.

1. Reply: We believe that the HVAC systems are no doubt for maintaining good indoor air quality and the HEPA filters can also help. (Page 29, line 412-415).

2.Comment: In the discussion read and quote the articles.

Gola m. capolongo S. et al.

Indoor air in healing environments: Monitoring chemical pollution in inpatient rooms. Facilities, 2019, 37(9-10), pp. 600–623.

2. Reply: Thank you very much for your recommendation. These articles on chemical pollution and energy efficiency complement our research. This article has been quoted on page 31, lines 455-456.

3. Comment: Existing guidelines for indoor air quality: The case study of hospital environments

SpringerBriefs in Public Health, 2017, (9783319491592), pp. 13–26

3. Reply: Thank you very much for your recommendation. This article has been quoted on page 3, lines 62-64

4. Comment: Indoor Air Quality in Inpatient Environments: A Systematic Review on Factors that Influence Chemical Pollution in Inpatient Wards

Journal of Healthcare Engineering, 2019, 2019, 8358306.

4. Reply: Thank you very much for your recommendation. This article has been quoted on page 31, lines 466-467.

5. Comment: The Dichotomy between Indoor Air Quality and Energy Efficiency in Light of the Onset of the COVID-19 Pandemic

Atmosphere 2021, 12, 791. https://doi.org/10.3390/atmos12060791

5. Reply: Thank you very much for your recommendation. This article has been quoted on page 32, lines 478-479.

Thank you again for your positive comments and valuable suggestions to improve the quality of our manuscript.

Response to Reviewer #2

1. Comment: Some paragraphs were founded on critical grammar errors. 

1. Reply: Thank you very much for pointing out our problem. We have tried our best to improve the manuscript and made some modification to the manuscript. These changes will not influence the content and framework of the paper. And here we did not list the changes but marked in red in file labeled 'Revised Manuscript with Track Changes'. We appreciate for your warm work earnestly and hope that the correction will meet with approval. 

We have made changes to the syntax of many sentences, for example:

Original: Databases of Embase, PubMed, Cochrane Library, MEDLINE, and Web of Science were searched from 1st January 2000 to 31th December 2020.

Revised: Databases of Embase, PubMed, Cochrane Library, MEDLINE, and Web of Science were searched from 1st January 2000 to 31st December 2020 (Page 1, line 17-18).

Original: However, the coronavirus disease 2019 (COVID-19) pandemic has raised people’s concern that HVAC systems may increase the risk of airborne diseases if they are not well designed or managed properly.

Revised: However, the coronavirus disease 2019 (COVID-19) pandemic has raised concerns that HVAC systems may increase the risk of airborne diseases if not well designed or properly managed (Page 3, line 62-64).

Original: After conversions, there were still multiple data with consistent classification, for example, some articles carried out the comparison between working state and non-working state [28] or the comparison among different kinds of operating [29].

Revised: After the conversions, there were still multiple data with consistent classification. For example, some articles compared working and non-working states [28] or compared different medical operation procedures [29] (Page 7, line 152-154).

Original: In the conditions using HEPA filters (bacterial count: 36.90 ± 40.06 CFU/m3; fungal count: 9.46 ± 9.63 CFU/m3), lower mean bioaerosol concentrations were obtained than in the conditions using other filters (bacterial count: 57.83 ± 85.06 CFU/m3; fungal count: 12.21 ± 14.22 CFU/m3).

Revised: Conditions wherein HEPA filters were used (bacterial count: 36.90 ± 40.06 CFU/m3; fungal count: 9.46 ± 9.63 CFU/m3) showed lower mean bioaerosol concentrations than those wherein other filters were used (bacterial count: 57.83 ± 85.06 CFU/m3; fungal count: 12.21 ± 14.22 CFU/m3) (Page 19-20, line 255-258).

Original: We usually think that in restricted areas of hospitals, the concentration of microbial bioaerosols should be lower because of more stringent management and disinfection measures.

Revised: It is typically thought that the concentration of microbial bioaerosols should be lower in restricted areas of hospitals because of more stringent management and disinfection measures (Page 25, line 326-327).

Original: These four studies are part of the difference in the sampling area in the operating room.

Revised: These four studies provide information on the differences in the sample areas of operating rooms (Page 28, line 386-387). 

2. Comment: Therefore, need to proofread one native researcher and provide the proofreading certificate supplementary file, and sent the revised version for peer review.

2. Reply: In the revised manuscript we have employed an English-language editing service, Editage to polish our wording. And a language certificate supplementary file has been uploaded (Certificate_of_editing-BSEQN_1_5_l3xzfkt2-g.pdf). We hope that the polished manuscript meets the publication requirements. We would like to express our great appreciation to you for your efforts spent on our manuscript.

---

## [Decision Letter · Decision Letter 1]

2 Nov 2021

A systematic review and meta-analysis of indoor bioaerosols in hospitals: the influence of heating, ventilation and air conditioning

PONE-D-21-19989R1

Dear,

We’re pleased to inform you that your manuscript has been judged scientifically suitable for publication and will be formally accepted for publication once it meets all outstanding technical requirements.

Kind regards,

Muhammad Shahzad Aslam, Ph.D.,M.Phil., Pharm-D

Academic Editor

PLOS ONE

Additional Editor Comments (optional):

Reviewers' comments:

Reviewer's Responses to Questions

**Comments to the Author**

1. If the authors have adequately addressed your comments raised in a previous round of review and you feel that this manuscript is now acceptable for publication, you may indicate that here to bypass the “Comments to the Author” section, enter your conflict of interest statement in the “Confidential to Editor” section, and submit your "Accept" recommendation.

Reviewer #1: All comments have been addressed

Reviewer #2: All comments have been addressed

2. Is the manuscript technically sound, and do the data support the conclusions?

Reviewer #1: Yes

Reviewer #2: Yes

3. Has the statistical analysis been performed appropriately and rigorously? 

Reviewer #1: Yes

Reviewer #2: Yes

4. Have the authors made all data underlying the findings in their manuscript fully available?

Reviewer #1: Yes

Reviewer #2: Yes

5. Is the manuscript presented in an intelligible fashion and written in standard English?

Reviewer #1: Yes

Reviewer #2: Yes

6. Review Comments to the Author

Reviewer #1: (No Response)

Reviewer #2: (No Response)

7. PLOS authors have the option to publish the peer review history of their article (what does this mean?). If published, this will include your full peer review and any attached files.

Reviewer #1: No

Reviewer #2: **Yes: **Dr. Yun Jin Kim, Ph.D

---

## [Editor Report · Acceptance letter]

7 Dec 2021

PONE-D-21-19989R1 

A systematic review and meta-analysis of indoor bioaerosols in hospitals: the influence of heating, ventilation, and air conditioning 

Dear Dr. Ji:

I'm pleased to inform you that your manuscript has been deemed suitable for publication in PLOS ONE. Congratulations! Your manuscript is now with our production department. 

Kind regards, 

on behalf of

Dr. Muhammad Shahzad Aslam 

Academic Editor

PLOS ONE